# Analysis of very-high surface area 3D-printed media in a moving bed biofilm reactor for wastewater treatment

Gabriel Proano-Pena[1]☯, Andres L. Carrano[2]☯, David M. Blersch[3]☯*

**1** Universidad Espíritu Santo, Guayaquil, Ecuador, **2** Department of Mechanical Engineering, School of Engineering, Fairfield University, Fairfield, Connecticut, United States of America, **3** Biosystems Engineering Department, Auburn University, Auburn, Alabama, United States of America

☯ These authors contributed equally to this work.
* dblersch@auburn.edu

**Data Availability Statement:** All data files are publicly available from the Digital Commons at Georgia Southern University (DOI 10.20429/data. 2020.1).

## Abstract

Moving Bed Biofilm Reactors (MBBRs) can efficiently treat wastewater by incorporating suspended biocarriers that provide attachment surfaces for active microorganisms. The performance of MBBRs for wastewater treatment is, among other factors, contingent upon the characteristics of the surface area of the biocarriers. Thus, novel biocarrier topology designs can potentially increase MBBR performance in a significant manner. The goal of this work is to assess the performance of 3-D-printed biofilter media biocarriers with varying surface area designs for use in nitrifying MBBRs for wastewater treatment. Mathematical models, rendering, and 3D printing were used to design and fabricate gyroid-shaped biocarriers with a high degree of complexity at three different levels of specific surface area (SSA), generally providing greater specific surface areas than currently available commercial designs. The biocarriers were inoculated with a nitrifying bacteria community, and tested in a series of batch reactors for ammonia conversion to nitrate, in three different experimental configurations: constant fill ratio, constant total surface area, and constant biocarrier media count. Results showed that large and medium SSA gyroid biocarriers delivered the best ammonia conversion performance of all designs, and significantly better than that of a standard commercial design. The percentage of ammonia nitrogen conversion at 8 hours for the best performing biocarrier design was: 99.33% (large SSA gyroid, constant fill ratio), 94.74% (medium SSA gyroid, constant total surface area), and 92.73% (large SSA gyroid, constant biocarrier media count). Additionally, it is shown that the ammonia conversion performance was correlated to the specific surface area of the biocarrier, with the greatest rates of ammonia conversion (99.33%) and nitrate production (2.7 mg/L) for manufactured gyroid biocarriers with a specific surface area greater than 1980.5 $m^2/m^3$. The results suggest that the performance of commercial MBBRs for wastewater treatment can be greatly improved by manipulation of media design through topology optimization.

**Funding:** This work was partially supported by Grant Number SU836122, from the U.S. Environmental Protection Agency (www.epa.gov) awarded to DMB and ALC. Material in-kind support provided to DMB by the Alabama Agricultural Experiment Station of Auburn University (aaes. auburn.edu). Partial support was from the Ecuadorian scholarship funded by the Secretaria de Educación Superior, Ciencia y Tecnología e Innovación-Senescyt (www.educacionsuperior. gob.ec), awarded to GPP. The funders had no role in study design, data collection and analysis, decision to publish, or preparation of the manuscript.

**Competing interests:** The authors have declared that no competing interests exist.

## Introduction

Moving Bed Biofilm Reactors (MBBRs) are used for nutrient removal and recovery applications in secondary and tertiary levels of wastewater treatment. These applications include pharmaceutical wastewater [1, 2], petroleum contaminated waters [3, 4], pulp and paper industry waste streams [5, 6], piggery wastewater treatment [7], and sustainable nutrient remediation in municipal wastewater treatment [8]. For applications with high nitrogen concentrations, MBBRs are commonly used for nitrification, the aerobic microbial bioconversion of ammonia nitrogen to nitrate nitrogen. Air supply injects oxygen to enhance nitrogen oxidation, while at the same time achieving mixing of the treated water and biocarrier media that carry the nitrifying microbial community [9, 10]. The biocarrier media in MBBRs provide a surface on which a functional microbial biofilm attaches, and mixing in the reactor replenishes nutrient-rich wastewater to the active biofilm surface on each biocarrier. The rate of reaction, determining the overall performance of the reactor, is determined in part by the surface area characteristics of the biocarrier media, including the media size, shape, surface area per unit volume (the specific surface area, or SSA), and the fill ratio in the reactor vessel [9, 11].

An approach to improve the treatment process efficiency of a MBBR may consider the role of the biocarrier media. For example, by increasing the media fill ratio (the volume of biocarriers relative to the total volume in the container vessel), the total surface area available in the reactor is increased, which should improve the reaction yields. Another alternative is to increase the SSA of the biocarrier (the ratio of the total surface area of the biocarrier to its own volume), usually attained by increasing the geometric complexity and features of the surface. However, the optimization of the biocarrier SSA is critical to the performance of MBBR technology, as it determines the functional characteristics of the biofilm through defining mass transfer characteristics, biofilm thickness, and biofilm attachment strength. For example, in nitrification MBBRs, where the microbial biofilm on the carriers performs the ammonia oxidation, excess biofilm growth limits oxygen mass transfer to deeper biofilm layers, leading to biofilm detachment through sloughing from the surface of the biocarrier [12] and consequently reducing nutrient removal performance. In this regard, previous research efforts [13] attempted to develop biocarrier geometries that control the biofilm thickness and thus the ammonia oxidation process. Otherwise, the lack of biofilm thickness control may transform the MBBR design into a hybrid unit (MBBR and suspended sludge) [14], which in turn affects the overall performance and operation. Improving on the design of biocarrier geometry is therefore seen as critical for maximizing the performance of the MBBRs and, consequentially, its economic viability.

Several research efforts have been directed at exploring variations on the design of the biocarrier, with the goal of improving the performance in MBBRs through the amount of bacterial biofilm retained [15]. These variations include shapes, sizes, and materials that can be varied to affect biocarrier performance [16–19]. Shape and size have been shown to control the oxygen mass transfer into the attached biofilm, demonstrating its key influence to shorten the start-up period on MBBRs [19]. Avoiding media clogging through excess biofilm is also an important consideration for the geometry of the biocarrier, and the effect of geometry on biofilm thickness and water flow regime has been shown to be a major factor [20]. Typically, any improvement in reactor performance is limited by the balance between available surface area per unit volume and topographical complexity, itself limited by the manufacturability of more complex designs.

Among the potential fabrication technologies for manufacturing artificial biocarriers, additive manufacturing (AM) is an emerging technology that provides the capability to fabricate biocarriers with high levels of geometric complexity. The feasibility of achieving microbial

colonization on 3D printed substrata has already been previously determined [21, 22], where AM was successfully used to produce both cubic gyroids and replicates of commercial carriers [22] that showed biofilm development in phototrophic biofilm reactors. Additional research demonstrated the effect of specialized topographic surfaces fabricated using AM on the species composition [23], colonization patterns [24], and functional growth and productivity [25] of phototrophic microbial biofilms. Other studies reported the fabrication of three fullerene-type microbial biocarriers using Laser Selective Sintering (SLS) technology and evaluated biofilm growth performance [26]. Under simulated sewage wastewater conditions in a sequencing batch reactor, the 3D-printed biocarriers developed a thicker biofilm than that found on the outer regions of the conventional Kaldnes K3 media, but developed thinner biofilms compared to those found in the inner walls. Tang et al. [2] used stereolithography technology (SLA) to fabricate a semi-suspended spindle-shaped biocarrier with synthetic material (isocyanate and polyhydric alcohols), which demonstrated the effects of shaped microhabitats on biofilm growth and microbial community diversity [2]. In preliminary work on the topic, Elliott et al. [27] demonstrated the manufacture of 3D-printed gyroid-based biocarriers using material jetting technologies and established the feasibility of implementing such technology for nitrification MBBRs [27].

To date, very little research has focused on the evaluation of novel 3D printed carriers in nitrifying MBBRs to reduce ammonia from wastewater and evaluate conversion rates as a function of the SSA or the fill ratio of biocarrier media. The 3D printed carriers reported in previous research [27] have the potential to accomplish faster nutrient conversion rates due to the large, mathematically modulated SSA. Additionally, gyroid-type carriers have geometric complexity that can define local hydrodynamics that are beneficial to biofilm formation rates and thus improve start-up times for MBBRs [28]. Given the fidelity of material jetting technology, and the easy scaling of gyroid geometry through parameter manipulation, different levels of void space can be designed in such a way that an optimal value of SSA can be determined that maximizes ammonia removal rates. Thus, the main objective of this research is to evaluate the performance of 3D printed gyroid biocarriers at different scaling parameters, SSA, and fill ratio in nitrification of wastewater in a MBBR.

## Materials and methods

### Biocarrier design and fabrication

Carriers were fabricated with additive manufacturing (i.e., 3D printing) technologies to produce complex geometries that span the limits on requirements for bacterial development in moving beds. Such requirements include a high SSA, an optimal void size (resulting in minimum clogging) and sufficient topographical sheltering (to protect bacterial biofilm from premature sloughing).

In this effort, a gyroid model was used to design the surface. This surface is generated from a mathematical equation with terms of sine and cosine functions (Eq 1) of an infinitely connected periodic minimal surface [29]. The equation model allows for manipulation of void size and overall surface area per unit volume by scaling the periodicity of the variables in the function.

$$\sin x * \cos y + \sin y * \cos z + \sin z * \cos x = 0 \tag{1}$$

Out of the infinite possible parameter combinations, three gyroid biocarrier geometries were designed based on previous work [27] as nominal spheres with 20 mm diameter, representing different configurations of SSA (large, medium, and small). The performance of these in nitrification MBBR was to be experimentally compared against a control group of Kaldnes

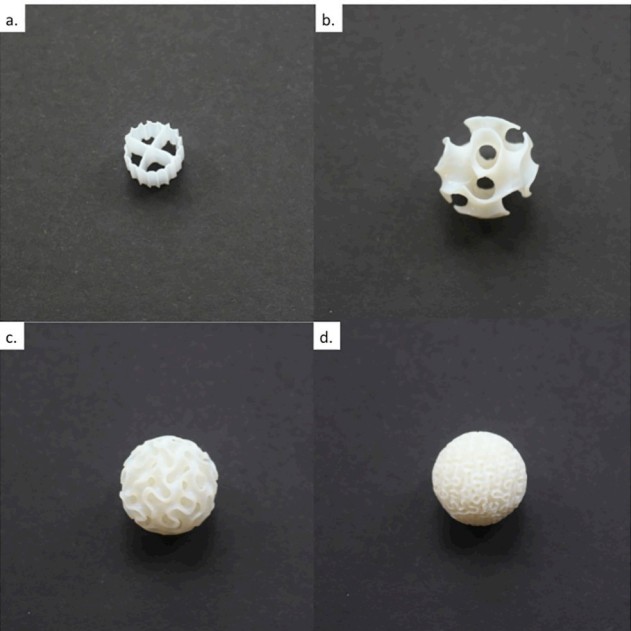

**Fig 1. Images of commercial (K1) and fabricated (gyroid) biocarrier designs examined in this research.** (a) Kaldnes K1; (b) small specific surface area gyroid; (c) medium specific surface area gyroid; (d) large specific surface area gyroid.

K1 media, a commonly-used commercial biocarrier design. The three gyroid designs were tessellated, converted into STL file format, and fabricated using an Objet-30 machine (Stratasys® Ltd., Eden Prairie, Minnesota), which uses a layer thickness of 28 μm of UV-light cured acrylic polymer to build the objects.

The SSA of the carriers was obtained with the aid of Netfabb®, an additive manufacturing software that calculates spatial information on the 3D models, including the total surface area, the true volume, and the volume of the work envelope of the carrier. True volume is defined as the consumed material (expressed as volume units) required to fabricate a single biocarrier object. The work envelope is the minimum closed surface that encompasses the entire geometry of the biocarrier. For the gyroid-based topologies used in this study, it consists of the minimum sphere that would theoretically wrap or enclose the entire biocarrier. For the Kaldnes K1 topology, it consists of the minimum cylinder that would encompass the entire biocarrier. The determination of the unit SSA for the gyroid biocarrier is the ratio of the surface area to the work envelope volume.

In a similar way, the unit SSA for the Kaldnes K1 commercial carriers was estimated, shown to be greater than the nominal value of 500 $m^2/m^3$ reported both in commercial and scientific literature [10]. To estimate the entire surface area of the commercial biocarrier, a reverse engineering approach was used, where the dimensions of the Kaldnes K1 biocarrier features were measured and the part was rebuilt via computer model using Solidworks® (Dassault Systèmes SolidWorks Corp., Waltham, Massachusetts). The file was then imported into Netfabb® and the information regarding its surface area retrieved. Images of example specimens of all biocarrier designs examined in this research are shown in Fig 1. Table 1 depicts the geometric parameters and material properties for all biocarrier designs that were used in this study.

**Table 1. Biocarrier designs and their geometric and material characteristics.**

| Name | Diameter (mm) | Surface area (m$^2$) | Volume of Work envelope (m$^3$) | Specific Surface Area (SSA)* (m$^2$/m$^3$) | Material density (kg/m$^3$) |
|---|---|---|---|---|---|
| Small SSA gyroid | 20.00 | 21.94 x10$^{-04}$ | 4.189 x 10$^{-6}$ | 523.8 | 1,033 |
| Medium SSA gyroid | 20.00 | 42.42 x10$^{-04}$ | 4.189 x 10$^{-6}$ | 1013 | 1,033 |
| Large SSA gyroid | 20.00 | 82.96 x10$^{-04}$ | 4.189 x 10$^{-6}$ | 1981 | 1,033 |
| Kaldnes K1 | 10.39 | 9.44 x10$^{-04}$ | 6.215 x 10$^{-7}$ | 1519 | 950** |

*Calculated specific surface area from design rendering models.

**Referenced material density from [30].

## Biofilter reactor design and setup

The experiments were conducted in a bench-scale Moving Bed Biofilm Sequencing Batch Reactor (MBBSBR) system, a sub-category of the broader category of MBBR. The experimental apparatus comprised two identical systems, each containing 6 reactors (Fig 2). Each reactor had a volume capacity for treating 1.3 liters of synthetic wastewater, and these were placed in a water bath to maintain constant temperature of 30˚C [30].

Water and environmental conditions were controlled so that the nitrification processes were the main biological mechanisms to take place. The reactors were built by using commercial 2L jars (height: 15 cm) to which an 11-cm aeration ring and sampling ports were incorporated. Aeration was necessary to provide aerobic conditions suitable for the nitrifying bacteria, and to provide motion and agitation to the carriers. To maintain a concentration of dissolved oxygen adequate for aerobic conditions to support nitrification (5 ppm, as in [10]), the air supply pressure for the bioreactors was kept constantly at an operational level of 1 psi, monitored with an analog pressure gage. An air relief and check valve was designed into the lid of each reactor jar to maintain constant pressure. Samples of water were obtained through a sampling port in each lid. When sampling was required, the air supply was stopped and the contents of the reactor were allowed to settle for 30 minutes; then the port plug was removed, and the sample was extracted with a 100 mL syringe.

The performance of the biocarriers was assessed by monitoring the concentrations of total ammonia and nitrate in a prepared solution of synthetic wastewater. The recipe and procedure for the preparation of the synthetic wastewater is based on the presence of dissolved ammonia supplied by ammonium chloride at a concentration of 10 mg $NH_4Cl$ L$^{-1}$. The theoretical minimum volume required in the commercial (Kaldnes K1) carriers for complete removal of ammonia within an 8-hour treatment process was estimated at 1.86 mL. This was supported by design calculations [31–33] for dissolved oxygen conditions within the range of 5–6 mg/L and an average temperature of 30˚C [10].

## Wastewater media preparation and analysis

The preparation of the synthetic wastewater media was based on procedures for assessing aquaculture systems found in literature [34]. The media recipe involved 17L of dechlorinated water, into which was dissolved ammonium chloride (170 mg), calcium carbonate (340 mg) and sodium bicarbonate (595 mg), plus trace elements necessary for nitrification provided by a low concentration (0.2% solution, 34 ml) marine salt solution (Seachem, Inc., Madison, Ga) [35]. Excess alkalinity of 27 mg/L (as $CaCO_3$) was added by calcium carbonate (15 mg/L) and sodium bicarbonate (35 mg/L), providing a buffering capacity to maintain pH in the range of 7.0–7.5 throughout an 8-hour long nitrification process. In addition, stock water was prepared to replenish evaporative losses. The stock water was dechlorinated tap water conditioned with

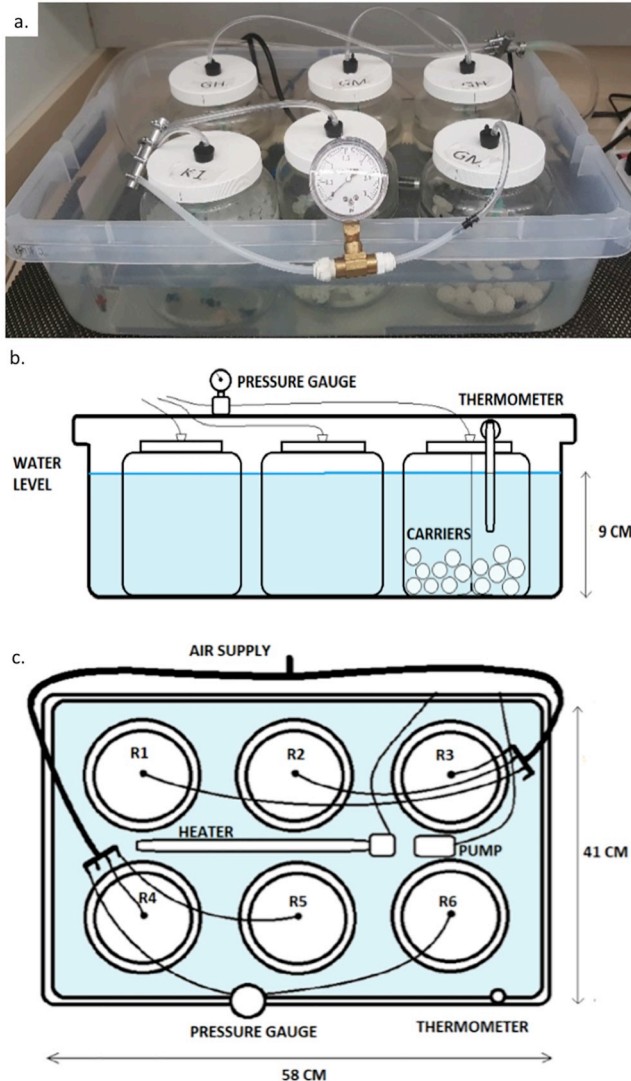

**Fig 2. Bench-scale MBBSBRs used for experimentation; (a) photograph of one system containing 6 reactors; (b) side-view schematic; (c) top-view schematic.**

off-the-shelf tap water conditioner (API, MARS Fishcare North America, Chalfont, PA) commonly used to dechlorinate aquarium waters. During the acclimatization period of multiple days, reactors ran continuously and evaporative losses were replenished with stock water every morning. Following preliminary operation and trials, synthetic wastewater was supplied using a high ammonia dilution of this media, resulting in an initial total ammonia nitrogen (TAN) concentration of 5 mg/L, typical for the lower range of concentrations for wastewaters.

Prior to introducing the carriers into the reactors for experimentation, they were exposed to bacterial inoculation and, hence, biofilm formation for approximately two months. Four buckets (each one per biocarrier type), each with 6 liters of synthetic wastewater and an air stone diffuser and a recirculation pump, were built for the purpose of inoculation.

A volume of 1.5 mL of Nitromax (Tropical Science Biolabs, Inc), an-off-the-shelf mixture of live *Nitrosomonas* and *Nitrobacter*, was added to each of the four buckets the first day

followed by 750 μL every other day. Daily evaporation losses were replenished with dechlorinated water.

Ammonia, ammonium, nitrite, nitrate, pH, chlorine, hardness and alkalinity were inspected daily with test strips (Tetra Test Strips, Spectrum Brands Pet, Blacksburg, Virginia) to monitor water quality status during the inoculation period. All parameters remained consistent within broad ranges indicated by test strips throughout the inoculation period. The pH remained within a constrained range (7.48–7.75) because of carbonate buffering; as such, ammonia and nitrite were below detection throughout inoculation and experiment periods.

After two months and once biofilm formation was achieved, the gyroid and Kaldnes K1 carriers were placed into their respective reactors with synthetic wastewater for acclimatization. Reactors were placed in the water baths at 30˚C nominal (measured at a 28.25±0.99˚C, n = 12), air diffusion was kept at 1 PSI on the pressure gauge, hence maintaining approximately 5–6 mg $O_2$/L in each reactor. A volume of 163 μL of Nitromax was added to each reactor every other day. This acclimatization process required daily inspection that was performed with test strips, digital thermometer, pH and DO probes to make sure that the conditions for nitrification were optimal. Two weeks after the start of the acclimatization process, the system started to cycle and to produce nitrates, as determined through daily inspection with test strips.

Concentrations of ammonia and nitrate were determined by colorimetry using a photometer (YSI 9500, Yellow Springs, Ohio) that measures the color intensity of the sample after the addition of extra reagents. The water quality was monitored using standard colorimetric methods. Ammonia nitrogen determination was based on an indophenol method using manufacturer reagents (Ammonia Reagent, VWR Catalog No. 55407–204). Nitrate nitrogen determination was performed with zinc-based methods (Nitratest Tablets, VWR Catalog No. 55407–152). Determinations were done on triplicate samples from each one of the six reactors and three times during each experimental trial: at the beginning, at the middle and at the end of an 8-hour run. Prior to each sampling process, the reactors were set to idle for 30 minutes to let the water stand, while the sampling process itself took 15 minutes. Because of this, by the time the samples were fully processed and stored for later analysis, they corresponded to the following time stamps: 00:45, 04:45, and 08:45 (hr:min). However, for practical purposes, results were reported to nominal values of 0 hours, 4 hours and 8 hours of treatment. The dissolved oxygen concentration, pH, and temperature measurements were obtained both at the beginning and at the end of every experimental trial, with pH remaining within the prescribed range (7.48–7.75), dissolved oxygen controlled (4.9–5.6 ppm), and temperature constrained (30˚C). The water dechlorination process to prepare the synthetic wastewater was done 24 hours prior to use. Most of the water analyses were performed immediately after sample collection.

## Experimental design and analysis of results

The overall guiding hypothesis for each experiment is that that a larger total surface area, whether provided on a media unit basis or spread across the bulk of the media, promotes faster rate of conversion of total ammonia nitrogen to nitrate nitrogen through the nitrification microbial process. In the research, three separate experiments were performed to measure the effect of biocarrier design on the performance of ammonia conversion to nitrate while maintaining constant across all reactors: (1) the biocarrier fill ratio and packing volume; (2) the total biocarrier surface area; and (3) the number (count) of biocarriers units (Fig 3). The packing volume, which is used to estimate the fill ratio, refers to the volume occupied by a

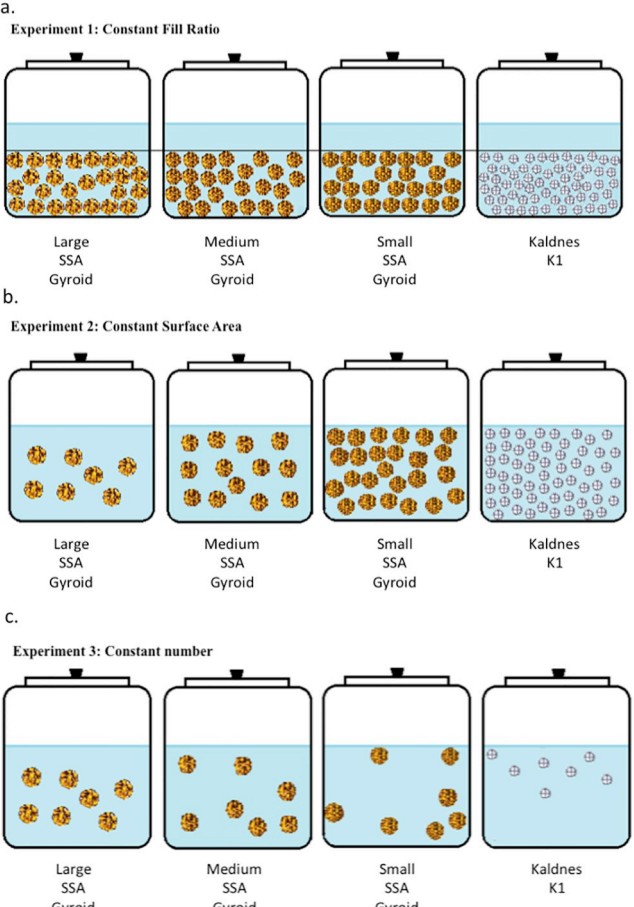

**Fig 3. Schematic of the experimental design for the three experiments.** (a) Experiment 1, testing constant biocarrier fill ratio; (b) Experiment 2, testing constant total biocarrier surface area; (c) Experiment 3, testing constant number of biocarriers. Diagram not to scale.

predetermined count of biocarrier elements and is dependent on the overall geometry and surface topology nesting characteristics, which would allow for a denser packing arrangement.

In every experiment, three trials were conducted. In every trial, each of the four different biocarrier designs were replicated three times thus employing twelve reactors, each treating synthetic ammonia-laden wastewater for 8-hour trials. In all experiments, sizing was determined based on calculations of minimum required biocarrier surface area for complete nitrification, as reported in [36, p.95] and validated through trial and error in preliminary trials (data not reported here). From preliminary trials it was identified that $2.00 \times 10^{-04}$ m$^3$ was a workable volume of packed biocarriers (for all types) that provided an observable mobility inside the reactor. Such volume selection consequently derived into the selection of the number of each carrier of each type for each of the experimental treatments described below.

For Experiment 1, the fill ratios were kept constant across trials (Table 2). Again, the fill ratio is the percentage of the reactor volume occupied by the bulk media when initially dry-loaded and packed tightly in the reactor. This required different quantities (count) of biocarrier elements depending upon the biocarrier design. The chosen fill ratio of 15% was consistent with values found in literature [12, 37, 38].

**Table 2. Parameters for Experiment 1 with constant fill ratio (15%).**

| Treatment/Biocarrier type | Surface area per biocarrier unit ($m^2$) | Count per reactor | Total surface area per reactor ($m^2$) |
|---|---|---|---|
| Kaldnes K1 | $9.44 \times 10^{-04}$ | 148 | 0.140 |
| Small SSA gyroid | $21.94 \times 10^{-04}$ | 25 | 0.055 |
| Medium SSA gyroid | $42.42 \times 10^{-04}$ | 25 | 0.106 |
| Large SSA gyroid | $82.96 \times 10^{-04}$ | 25 | 0.207 |

For Experiment 2, the total biocarrier surface area was kept constant across all trials (Table 3). The total biocarrier surface area is the summation of the available surface area of all biocarriers. Because of the different specific surface area for each type of biocarrier, a different number (count) of biocarriers was required for each trial unit. The total surface area per reactor of 0.055 $m^2$ was chosen based on the minimum amount of surface area for gyroids needed for a measurable rate of nitrification throughout the time of the trial, as determined through trial and error preliminary investigations and confirmed through repetition in other experiments (for example, Experiment 1, Table 2).

Finally, for Experiment 3, the total number (count) of carriers was kept constant across all trials (Table 4). Because of the different specific surface area for each type of biocarrier, the same count resulted in different total surface area and fill ratio per reactor for different biocarriers. The constant count value of 7 units of biocarriers in each reactor was chosen based on the minimum number of Kaldnes K1 biocarriers needed for a detectable amount of total ammonia conversion during the 8-hour trials, as determined through observations in trial and error preliminary investigations (data not included here).

For each experiment and trial, the experimental design was a full factorial with three replicate reactors (4 factors x 3 reps = 12 reactors/trial) and over three experimental trials for a total of 36 observations (9 per biocarrier treatment). The experimental design was conducted by blocking the reactors into two separate baths with six reactors each. The locations of the reactors within and between baths were randomly assigned in each trial and throughout the entire experiment to guard against uncontrolled factors. An analysis of homoscedasticity and a Kolmogorov-Smirnov normality test showed that the data do not follow the underlying

**Table 3. Parameters for Experiment 2 with constant total biocarrier surface area (0.055 $m^2$).**

| Treatment/Biocarrier Type | Packing Volume ($m^3$) | Surface Area per Biocarrier Unit ($m^2$) | Count per reactor | Fill Ratio per Reactor (%) |
|---|---|---|---|---|
| Kaldnes K1 | $6.91 \times 10^{-05}$ | $9.44 \times 10^{-04}$ | 58 | 5 |
| Small SSA gyroid | $2.00 \times 10^{-04}$ | $21.94 \times 10^{-04}$ | 25 | 15 |
| Medium SSA gyroid | $1.04 \times 10^{-04}$ | $42.42 \times 10^{-04}$ | 13 | 8 |
| Large SSA gyroid | $5.30 \times 10^{-05}$ | $82.96 \times 10^{-04}$ | 7 | 4 |

**Table 4. Parameters for Experiment 3 with constant total number (count) of biocarriers (7 biocarriers each).**

| Treatment/Biocarrier Type | Packing Volume ($m^3$) | Surface Area per Biocarrier Unit ($m^2$) | Total surface area per reactor ($m^2$) | Fill ratio per reactor (%) |
|---|---|---|---|---|
| Kaldnes K1 | $8.33 \times 10^{-06}$ | $9.44 \times 10^{-04}$ | 0.007 | <1 |
| Small SSA gyroid | $5.30 \times 10^{-05}$ | $21.94 \times 10^{-04}$ | 0.015 | 4 |
| Medium SSA gyroid | $5.30 \times 10^{-05}$ | $42.42 \times 10^{-04}$ | 0.030 | 4 |
| Large SSA gyroid | $5.30 \times 10^{-05}$ | $82.96 \times 10^{-04}$ | 0.055 | 4 |

assumptions for an Analysis of Variance. Consequently, the statistical analysis was conducted with a non-parametric Kruskal-Wallis statistical test on the medians. Additional analysis was performed with a Mood's Median test to compare the medians of pairwise samples to determine significance. Additionally, before each experimental trial, all carriers of the same type were mixed together in a bag and randomly selected and replaced into the reactors. All conditions were reset after each trial and new assignments of reactor locations and biocarrier sets were randomly conducted. For all experiments, water samples were taken regularly and analyzed for ammonia and nitrate nitrogen concentrations, for comparison across all treatments of nitrification performance. Ammonia concentration conversion percentage was calculated for each replicate, where the pooled ammonia nitrogen concentration at the 8th hour was deducted from the concentration at the start of the water treatment, and the difference divided by the starting concentration of ammonia nitrogen. These are reported and compared for all trials and all experiments.

## Results and discussion

### Experiment 1: Constant fill ratio of biocarriers

The results from Experiment 1, constant Fill Ratio of biocarriers, are presented in Fig 4a, with parameters expressed as pooled Total Ammonia Nitrogen (TAN) and Nitrate Nitrogen concentrations over time. Results show that gyroids with medium and large SSA convert significantly more ammonia than the commercial Kaldnes K1 and the gyroid with small SSA ($p < 0.001$). Two separate Mood's median tests (5% significance level) were performed on the total ammonia nitrogen and nitrate nitrogen. These tests could not differentiate between the performances of large SSA and medium SSA gyroids at 4 and 8 hours, so these were statistically found to provide the same performance. Similarly, from the same statistical test, the performances of the small SSA gyroid and the Kaldnes K1 were found to be statistically the same at 4 and 8 hours. The observed variation in the ammonia conversion performance when manipulating the SSA parameter reinforces previously reported findings [39] that SSA is an important parameter for MBBR design. Furthermore, these findings strongly suggest that SSA can be used as a proxy parameter when seeking to optimize MBBR performance. Also, the amounts of ammonia conversion by the medium and large SSA gyroid types were similar (close to 100% of ammonia removed from solution) with the same biocarrier volume (as expressed by fill ratio) in the reactors. This suggests that the subsidy of increased surface area is limited, preventing a continued increase in ammonia conversion rate despite the increased SSA. Previous studies suggest that a smaller biocarrier pore size results in slower rates of substrate and oxygen transport to inner regions inside the voids [28]. This might apply to the large SSA gyroid, which has the smallest pore sizes of the gyroid biocarriers, thus preventing an increase in nitrification performance proportional to the increase in SSA.

The observed trends in nitrate concentration were similar, where the gyroids with medium and large SSA increased nitrate concentrations at a greater rate than the small SSA gyroid and Kaldnes K1. By comparing the changes in concentration of ammonia nitrogen and nitrate concentration over time, a conservation of nitrogen was observed within the process. In all trials, the amount of ammonia nitrogen consumed matched the amount of nitrate nitrogen produced, confirming the nitrification performance of the biofilters.

Percentage change for total ammonia nitrogen for all trials was calculated and is shown in Fig 5. In Experiment 1, the 8-hour percentage change of total ammonia nitrogen was nearly 100% conversion for the highest-performing treatments (Gyroids with Large and Medium SSA), whereas the small SSA gyroid and Kaldnes K1 media had significantly lower TAN change percentage (42% and 30%, respectively). The significantly higher TAN conversion

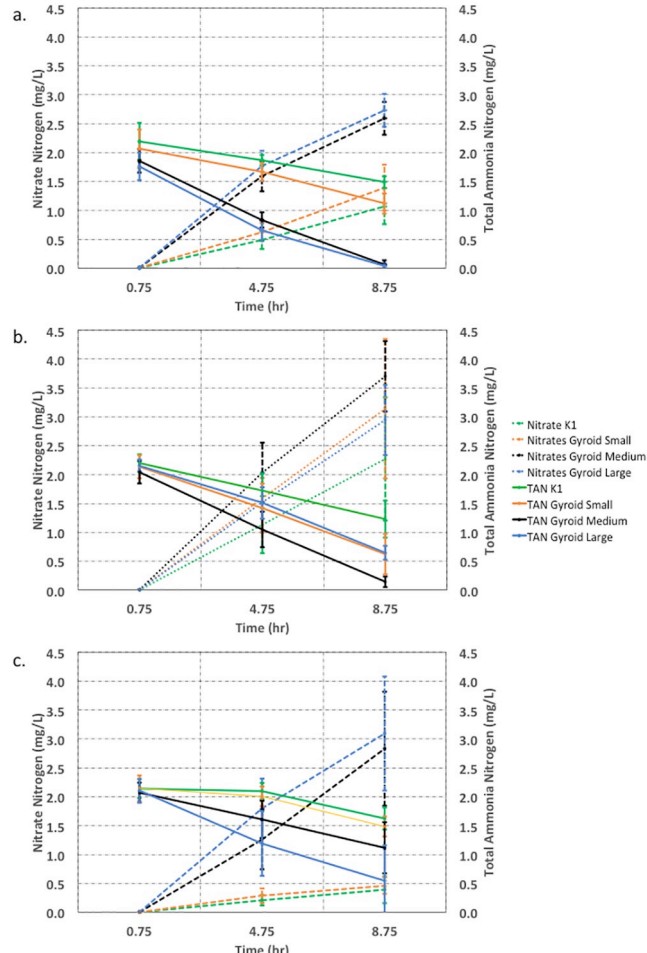

**Fig 4. Pooled results for total ammonia nitrogen and nitrate nitrogen concentrations over time for (a) Experiment 1, constant biocarrier fill ratio; (b) Experiment 2, constant biocarrier total surface area; (c) Experiment 3, constant biocarrier count.** Error bars represent standard deviation.

percentage correlates with the higher specific surface area, and thus total surface area, in the reactors with the medium and large SSA gyroid.

Results on the performance of the different media in Experiment 1 also help to understand the role of surface area conformation in establishing performance. From results of Experiment 1, it is readily observed that the percentage conversion of ammonia nitrogen correlates with the SSA per biocarrier type. Such relationship was identified between the treatments with small (low) and medium SSA gyroids. In such treatments, the amount of ammonia nitrogen converted by the medium SSA gyroids was approximately twice that converted by the small SSA gyroids, corresponding to the doubling of the SSA for medium gyroids (Fig 6). The performance in TAN conversion did not necessarily correlate with total surface area in the reactor, where the Kaldnes K1 media had the second largest total surface area but the lowest TAN percent change (Fig 6). The amount of ammonia nitrogen converted by Kaldnes K1 was not greater than the amount of ammonia nitrogen converted by the medium or small SSA gyroids, even though the SSA of the Kaldnes K1 ($1519 \ m^2/m^3$) is nominally larger than the other two ($1013$ and $524 \ m^2/m^3$ respectively). However, the amount of ammonia nitrogen converted by the Kaldnes K1 biocarrier was consistent with its reported surface area of $500 \ m^2/m^3$ as

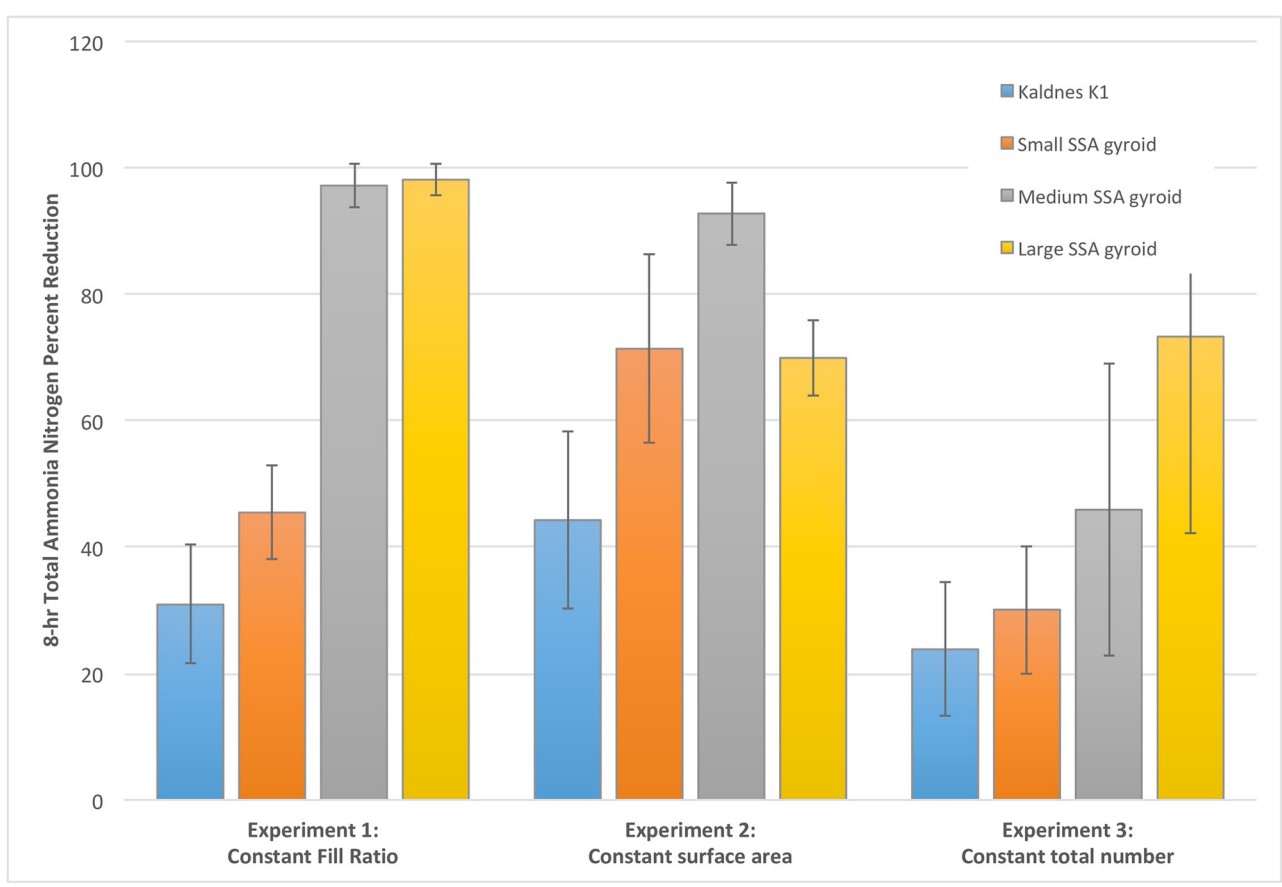

**Fig 5. 8-hour total ammonia nitrogen (TAN) percentage change for each biocarrier type in each experiment (error bars represent standard deviation).**

communicated by the manufacturer [40, 41]. This suggests that the performance of the gyroid media is enhanced beyond the subsidy of greater SSA, with this enhancement coming from materials considerations or particular characteristics of the gyroid shape.

## Experiment 2: Constant biocarrier total surface area

The results from Experiment 2, constant total surface area, are presented in Fig 4b, with parameters expressed as pooled total ammonia nitrogen (TAN) and nitrate nitrogen concentrations over time. Results showed a decrease in ammonia nitrogen concentrations over time for all trials (Fig 4b). The greatest rate of conversion of ammonia was seen for the medium SSA gyroids; the next greatest rates of conversion seen for large and small SSA gyroids; and the smallest rate of conversion observed for the Kaldnes K1 media. Results were significant ($p < 0.001$, Kruskal-Wallis test) for the differences in 8-hour ammonia nitrogen conversion between all gyroids and the Kaldnes K1 media. Similarly, nitrate nitrogen concentration increases followed the same pattern (Fig 4b), with the greatest change observed for the medium SSA gyroid and the least for the Kaldnes K1, with significance ($p < 0.001$). Two separate Mood's median tests (5% significance level) were performed on the total ammonia nitrogen and nitrate nitrogen. These tests could not differentiate between the performances of large SSA, medium SSA, and small SSA gyroids at 4 and 8 hours, so these were statistically found to provide the same performance.

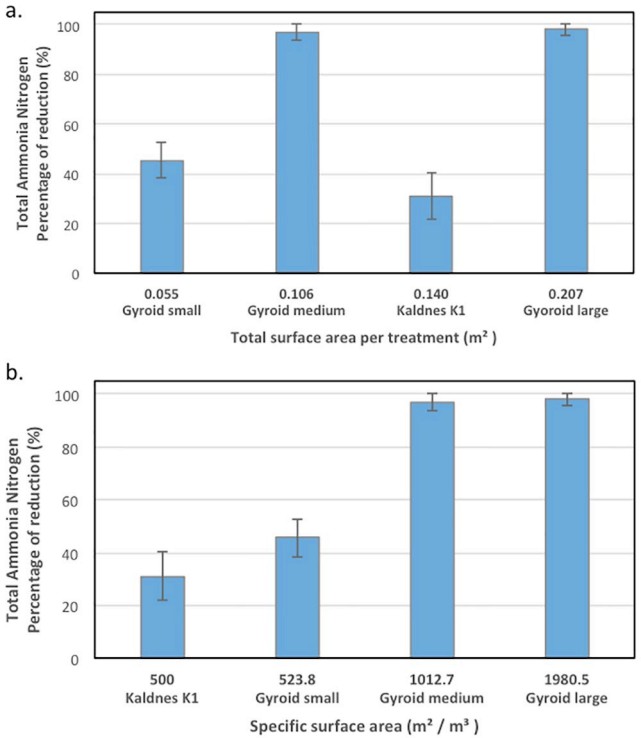

**Fig 6. Total ammonia conversion for each biocarrier as a function of (a) total surface area per treatment, and (b) specific surface area of the biocarrier, for Experiment 1, constant fill ratio.**

Percentage change of ammonia nitrogen concentration after 8 hours for Experiment 2 (Fig 5) shows a greater performance for all gyroids than Kaldnes K1 ($p<0.001$). It is important to highlight the performance of medium SSA gyroids, where almost 100% of ammonia nitrogen was converted within 8 hours, approaching the same conversion rate as the high SSA gyroids in Experiment 1 (0.40 mg/L/hour), even though the number of biocarriers was decreased (25 to 13) in these trials. A potential reason for this continued high performance is that, under the conditions in Experiment 2, the decrease in the number of biocarriers in the reactor allowed for increased agitation and increased mixing intensities, enhancing mass transfer of oxygen or substrate [40, 41]. Additionally, the high performance of the medium SSA gyroid, compared to others, suggests that an optimum configuration of surface area exists, given that the total surface area in all trials is the same. The lower performance the large SSA gyroid suggest the limitation of SSA on performance, likely as a result of mass transfer limitations on smaller features and pore sizes of the large SSA gyroid. This result suggests the perspective that SSA and total surface area of media in a reactor could be optimized for best performance through a combination of media SSA complexity and total count (fill ratio) of media in the reactor.

### Experiment 3: Constant biocarrier count

The results from Experiment 3, constant biocarrier count, are presented in Fig 4c, with parameters expressed as pooled total ammonia nitrogen (TAN) and nitrate nitrogen concentrations over time. As before, results showed a decrease in ammonia nitrogen concentration over time for all trials. The greatest rate of conversion of ammonia was seen for the large SSA gyroids; a

moderate rate of conversion seen for medium SSA gyroid; and the smallest rate of conversion observed for both the small SSA gyroid and Kaldnes K1 biocarriers. As before, results were significant (p<0.001, Kruskal-Wallis test) for differences in 8-hour ammonia concentration decrease between all gyroids and the Kaldnes K1 media. Similarly, nitrate nitrogen concentration increases followed the same pattern (Fig 4c), with the greatest change observed for the large SSA gyroid and the least for the Kaldnes K1, with significance (p<0.001). Two separate Mood's median tests (5% significance level) were performed on the total ammonia nitrogen and nitrate nitrogen. These tests could not differentiate between the performances of large SSA, medium SSA, and small SSA gyroids at 4 and 8 hours so these were statistically found to provide the same performance.

Percentage change of ammonia nitrogen concentration over 8 hours for Experiment 3 (Fig 5) shows a greater rate of total TAN concentration decrease for all gyroids than for Kaldnes K1 media ($p<0.001$). TAN percentage change was 71% for the large SSA gyroid, 45% for medium SSA gyroid, 30% for small SSA gyroid, and 22% for Kaldnes K1 biocarrier. The results show the subsidy of greater specific surface area of the gyroid media, where TAN percentage conversion performance increases with increasing specific surface area of the biocarrier media. All gyroid trials had a greater TAN conversion performance than the Kaldnes K1 media at the equivalent number. The TAN percentage change from gyroids was measurably greater than conventional biocarriers even when the biocarrier count was notably less than that compared to other experiments. This suggests that the performance of the gyroid biocarriers is enhanced by the high specific surface area. In addition, the amount of increases in performance for the medium and large SSA gyroids, despite the relatively small number of biocarriers in this experiment, suggests a subsidy in performance resulting from the high amount of internal surface area, a result directly related to the conformation of the topographical design of the biocarriers.

## Summary of all results

Overall, the manufactured gyroids of all SSA showed high performance in nitrification MBBR applications. In all experiments, the nitrification rate was directly related to the specific surface area of the biocarrier, and the cumulative nitrification was influenced by the total surface area in the reactor, accumulated by the combination of the specific surface area of the media and the total count of the individual media carriers. In all cases, even when reactors were normalized for total surface area available, total fill ratio, and total count, manufactured gyroid biocarriers had a greater nitrification performance than commercial Kaldnes K1 media. It is expected that the generally greater SSA with more shape conformation, providing a large amount of surface area that could protect an established active biofilm, contributes to the overall subsidy of performance. Optimization of the surface area complexity for the gyroids is necessary to understand the effects of long-term operation on biofilm accumulation, mass transport dynamics, and ultimately reactor-scale performance in operations. Additionally, performance of the biocarriers should be investigated for applications to wastewater streams, including municipal or aquaculture sources, where higher concentrations in organic carbon and nutrient concentrations differentially impact the accumulation of biofilm on the carriers themselves, and ultimately the performance of the bioreactor in any given application. Life span considerations are important, too, as the material and shape integrity of the biocarriers over long-term repeated use is unknown. The approach for manufacture and application of gyroid geometry as a biocarrier, however, is demonstrated through the significant performance increases observed in these experiments.

## Conclusions

The levels of geometrical complexity of the design and fabrication of biocarriers for water treatment processes were manipulated with the capabilities of additive manufacturing. Three mathematical designs with increasing specific surface area (SSA) were used to fabricate biocarriers and test their performance in three different scenarios: constant fill ratio, constant total surface area, and constant biocarrier media count. For all the scenarios, the gyroid media designs presenting large and medium SSA provide the best nitrification performances: that is, the lowest final concentration of pooled total ammonia nitrogen as well the highest concentration of pooled nitrate nitrogen. The best performing design for the scenario of constant fill ratio is the large SSA gyroid with 99.33% of ammonia nitrogen conversion at 8 hours. The best performing design for the scenario of constant total surface area is the medium SSA gyroid with 94.74% of ammonia nitrogen conversion at 8 hours. The best performing design for the scenario of constant biocarrier media count is the large SSA gyroid with 92.73% of ammonia nitrogen conversion at 8 hours. From a practitioner's standpoint, Experiment 1 perhaps provides the most relevant information for selection of biofilter media, as one of the most practical considerations in bioreactors loading conditions is the fill ratio in the tanks. In this regard, and with manufacturing considerations aside, the results indicate that either large or medium gyroid SSA would be the best choice. However, this study was focused on the early stages of nitrification and thus limited to an 8-hour period. Therefore, it ignores some of the steady-state operational issues (e.g. clogging, biofilm build-up, sloughing, etc.) of bioreactors that would likely give a functional advantage to a design with larger pore openings while maintaining optimal performance (i.e. medium SSA gyroid). Additionally, results from Experiments 2 and 3 indicate that further optimization on the geometry and its proxy SSA can be pursued. Future work will aim at this through the use of statistical response surface methodologies over a longer period of bioreactor performance.

## Author Contributions

**Conceptualization:** Gabriel Proano-Pena, Andres L. Carrano, David M. Blersch.

**Data curation:** Gabriel Proano-Pena, Andres L. Carrano.

**Funding acquisition:** Andres L. Carrano, David M. Blersch.

**Investigation:** Gabriel Proano-Pena.

**Methodology:** Gabriel Proano-Pena.

**Project administration:** Andres L. Carrano, David M. Blersch.

**Resources:** Andres L. Carrano, David M. Blersch.

**Supervision:** Andres L. Carrano, David M. Blersch.

**Validation:** Gabriel Proano-Pena.

**Visualization:** Gabriel Proano-Pena.

**Writing – original draft:** Gabriel Proano-Pena.

**Writing – review & editing:** Gabriel Proano-Pena, Andres L. Carrano, David M. Blersch.

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
