## [Decision Letter · Decision Letter 0]

4 Jun 2020

PONE-D-20-13545

Analysis of Very-High Surface Area 3D-printed Media in a Moving Bed Biofilm Reactor for Wastewater Treatment

PLOS ONE

Dear Dr. Blersch,

Thank you for submitting your manuscript to PLOS ONE. After careful consideration, we feel that it has merit but does not fully meet PLOS ONE’s publication criteria as it currently stands. Therefore, we invite you to submit a revised version of the manuscript that addresses the points raised during the review process.

We look forward to receiving your revised manuscript.

Kind regards,

Leonidas Matsakas

Academic Editor

PLOS ONE

Journal Requirements:

Reviewers' comments:

Reviewer's Responses to Questions

**Comments to the Author**

1. Is the manuscript technically sound, and do the data support the conclusions?

Reviewer #1: Partly

Reviewer #2: Yes

2. Has the statistical analysis been performed appropriately and rigorously? 

Reviewer #1: N/A

Reviewer #2: No

3. Have the authors made all data underlying the findings in their manuscript fully available?

Reviewer #1: No

Reviewer #2: Yes

4. Is the manuscript presented in an intelligible fashion and written in standard English?

Reviewer #1: Yes

Reviewer #2: Yes

5. Review Comments to the Author

Reviewer #1: PONE-D-20-13545

Analysis of Very-High Surface Area 3D-printed Media in a Moving Bed Biofilm Reactor for Wastewater Treatment

The study is about the wastewater treatment specific to ammonia/TAN using a high surface area 3D fabricate gyroid-shaped bio-carrier. Treatment performance was compared with the market available commercial bio-carrier. Wastewater treatment results in terms of ammonia removal are interesting with 3D fabricated gyroid-shaped bio-carriers compared to conventional bio-carrier. However, the experiments were carried out using synthetic wastewater. The actual performance/potential of bio carriers can be examined with real field wastewater even at the lab scale. The discussion on removal is weak, not focused the treatment efficiency.

Comments

• The photograph showed for commercial bio-carrier is actually not such bad as shown here, it can be accepted that 3D-Printed bio carrier is better but portraying such images for commercial bio-carrier is not acceptable.

• Real field wastewater treatment is always advisable to evaluate the potential of designed bio carriers. The process may be examined with real field wastewater if possible.

• What is the life of designed bio carriers and their interaction with microbes?

• Quality of graphs are very poor to read the contents.

• Why the initial concentration of nitrates and TAN was chosen below 5 mg/L as wastewater typically contains more than 20 mg/L.

• Any figure can be produced with biofilm formation on designed and conventional bio carrier.

• Ammonia, ammonium, nitrite, nitrate, pH, chlorine, hardness, and alkalinity were also inspected but values were not mentioned in the study.

• Line 303: it cannot be claimed that SSA is the main parameter for maximizing the performance of MBBRs as many other factors influence the whole treatment process.

• Ammonia removal close to 100% might be due to the adaption of microbes at low concentration of ammonia being fed in the bioreactor- the case might be opposite with real field wastewater or when the concentration of ammonia is more than 5 mg/L.

• Any relation found for nitrate production and TAN removal, as nitrate production is almost the same whereas trend for TAN removal is quite opposite.

Reviewer #2: Abstract: Some data must be provided to support your statements.

Page 3, lines 41-44: Introduce the following current reference regarding the applications of MBBR in order to complement the literature cited: “Moving bed biofilm reactor as an alternative wastewater treatment process for nutrient removal and recovery in the circular economy model”.(https://doi.org/10.1016/j.biortech.2019.122631).

Page 3, line 53: Include the following reference in relation to the filling ratio: “Effect of the filling ratio, MLSS, hydraulic retention time, and temperature on the behavior of the hybrid biomass in a hybrid moving bed membrane bioreactor plant to treat urban wastewater”. (10.1061/(ASCE)EE.1943-7870.0000939)”.

Page 3, line 60: Correct “MMBR”.

Page 6, line 128: you mention reference 23, but in introduction you say that this kind of design had not been carried out previously for MBBR. Could you justify this contradictory information?

Figure 1: Explain why you introduce this figure in the paper.

Page 7, line 154: Why do you indicate “Table 1” in bold?

Table 1: some lines of the table are missing. Specify the meaning of SSA and include the abbreviation in brackets. Indicate the reference where you have obtained the value of SSA and density for K1 carrier from. The meaning of “work envelope” must be explained.

Figures 2 - 6: They have not enough quality. Please improve them.

Page 8, line 174: 5 ppm for dissolved oxygen is excessive.

Section “Wastewater media preparation” is too long. You must divide it into more sections. Furthermore, some descriptions of analytical methods can be summarized or included in Supplementary Information.

Table 2: remove it and indicate the volume in the text (since the concentrations are given).

Page 10, line 221: What does “uL” mean?

Page 12, line 267: Justify the choice of 0.055 m2. It is not clear in the text.

Table 4: Explain the meaning of “packing volume” in the text in order to understand the information supplied in this table.

Page 12, line 276: justify with some reference the number of 7 carriers.

Section “Results and Discussion” must be divided in several sub-sections according to the three experiments carried out.

Page 13, line 300: do not use capital letters and use “total ammonia nitrogen” without “-N”.

Page 13, line 302: it is not logical to give a p-value with three significant numbers. Revise p-values in manuscript and introduce the statistical method in “Materials and Methods”. I can not revise the results from Figure 4 as I can not see them (poor quality of the figure). Please, do not use abbreviations in X or Y axis.

When you say “ammonia removed”, what do you refer to? In your systems, there is no an anoxic zone or anoxic time. Explain it better in order to get a better comprehension for potential readers. I suppose that you consider the transformation of ammonium nitrogen into nitrate nitrogen (nitrification only). Specify clearly it (“remove” is confusing). Additionally, explain why you analyze both parameters TAN and nitrate concentration.

Page 15, lines 345-349: compare with the values of protected surface area for biofilm growth corresponding to your biocarriers.

Figure 6: it is not easily interpretable. Moreover, use superscripts.

Page 16, lines 394-395: Justify what you say.

Pages 16-17, lines 396-400: Explain it with more detail.

Conclusions: this section must be summarized, you must give conclusions without repeating information from Materials and Methods for example. You can number the main conclusions for each experiment carried out. Include some data to support your conclusions. Explain the last paragraph: you state 1013 m2/m3 as the most suitable carrier, which carrier does this value correspond to? Justify this last paragraph.

6. PLOS authors have the option to publish the peer review history of their article (what does this mean?). If published, this will include your full peer review and any attached files.

Reviewer #1: No

Reviewer #2: No

---

## [Author Response · Author response to Decision Letter 0]

13 Jul 2020

Response to Reviewers

We present the response to reviewers in detail in this communication. We thank the reviewers for their helpful and constructive comments that will ultimately improve this manuscript. We believe we have addressed all comments in this letter and/or in the manuscript. 

In this response letter, reviewer comments are shown in italicized font, followed by our response in standard text. The additions or edits of significance to the manuscript are in shown in bold text as part of the response. The location for the changes made to the document is also indicated within the response.

Reviewer #1: 

The study is about the wastewater treatment specific to ammonia/TAN using a high surface area 3D fabricate gyroid-shaped bio-carrier. Treatment performance was compared with the market available commercial bio-carrier. Wastewater treatment results in terms of ammonia removal are interesting with 3D fabricated gyroid-shaped bio-carriers compared to conventional bio-carrier. However, the experiments were carried out using synthetic wastewater. The actual performance/potential of bio carriers can be examined with real field wastewater even at the lab scale. The discussion on removal is weak, not focused the treatment efficiency.

Specific Comments

• The photograph showed for commercial bio-carrier is actually not such bad as shown here, it can be accepted that 3D-Printed bio carrier is better but portraying such images for commercial bio-carrier is not acceptable.

Thank you for your comment and observation on the figure. Indeed, the figure is of poor quality, and our random selection does not adequately reflect all the biocarrier designs examined in this research. We have produced a new image of the commercial bio-carrier, and added images of each of the fabricated biocarriers for an improved overview presentation of our research biocarriers. Along with this, we have updated the caption for Figure 1 (line 166-168, Manuscript Markup) to read as follows:

Figure 1: Images of commercial (K1) and fabricated (gyroid) biocarrier designs examined in this research. (a) Kaldnes K1; (b) small specific surface area gyroid; (c) medium specific surface area gyroid; (d) large specific surface area gyroid.

• Real field wastewater treatment is always advisable to evaluate the potential of designed bio carriers. The process may be examined with real field wastewater if possible.

We thank the reviewer for this comment. Indeed, additional value into understanding the performance of the novel biocarriers would be generated from investigating with real wastewater. We chose not to investigate this for this initial investigation, however, because of the desire to precisely control the chemistry of the input water to reduce experimental noise in a constrained experimental design. The potential variability in water chemistries, including variability in both nutrient and organic concentrations, of most wastewaters was undesirable for planning our initial investigations here. As such, our media was not intended to fully emulate municipal sourced wastewater, but more so a high-nutrient low organic carbon wastewater that might be found in aquaculture applications, where mixed bed bioreactor technologies are often used for nitrification. We strongly feel that future work on these biocarriers should include investigations with real wastewater sourced from multiple sources and of varying water quality and nutrient and organics concentration. We have added the following sentence (lines 474-478, Manuscript Markup) to indicate this:

Additionally, performance of the biocarriers should be investigated for applications to wastewater streams, including municipal or aquaculture sources, where variations in organic carbon and nutrient concentrations differentially impact the accumulation of biofilm on the carriers themselves, and ultimately the performance of the bioreactor in any given application.

• What is the life of designed bio carriers and their interaction with microbes?

We appreciate this comment as it relates to one of our main concerns and interests. We agree that life span of any biocarrier is an important operational consideration that would impact use, utility and potential commercial success of the biocarrier in any application. In this research, however, we are focused on the fundamental aspects of design of the shape and the performance aspects in the early initiation stages of biofilm development and application (i.e. the first 8 hours of operation). Further research is indeed required to determine the durability and lifespan aspects of these biocarriers.. To address this concern, we have added the following sentence (lines 478-479, Manuscript Markup) to our summary paragraph:

Life span considerations are important, too, as the material and shape integrity of the biocarriers over long-term repeated use is unknown.

• Quality of graphs are very poor to read the contents.

Thank you for pointing this out to us.. We apologize for the poor quality of the graphs, which is suspected to have occurred during file conversion upon upload of images. We have re-formatted all figure images with high resolution files, which is expected to improve the quality of all figures. 

• Why the initial concentration of nitrates and TAN was chosen below 5 mg/L as wastewater typically contains more than 20 mg/L.

Thank you for your comment. The reviewer is correct that wastewater often has TAN concentrations higher than 5 mg/L. We chose the concentration of TAN to reflect a low-end concentration, that is often typical of wastewaters from biofloc aquaculture production. The reviewer comment is important and formative, however, for future work regarding the optimization of performance for our biofilter media. We have added clarification of this in the Methods section (lines 214-217, Manuscript Markup):

Following preliminary operation and trials, synthetic wastewater was supplied using a high ammonia recipe (Table 2), resulting in an initial total ammonia nitrogen (TAN) concentration of 5 mg/L, typical for the lower range of concentrations for wastewaters.

Also, we have added this consideration to recommendations for future research (lines 474-478, Manuscript Markup):

Additionally, performance of the biocarriers should be investigated for applications to wastewater streams, including municipal or aquaculture sources, where higher concentrations in organic carbon and nutrient concentrations differentially impact the accumulation of biofilm on the carriers themselves, and ultimately the performance of the bioreactor in any given application.

• Any figure can be produced with biofilm formation on designed and conventional bio carrier.

Thank you for your comment. Although we confirmed the presence of biofilm on all biocarriers with microscopy, we do not have any images showing this, and chose to not report that investigation in this manuscript. 

• Ammonia, ammonium, nitrite, nitrate, pH, chlorine, hardness, and alkalinity were also inspected but values were not mentioned in the study.

Thank you for your comment. We will attempt to clarify. Most parameters mentioned here were monitored to ensure that they remained within prescribed ranges known to be favorable for nitrification. Many parameters (chlorine, hardness, alkalinity) were measured via colorimetric test strips, which provide a broad range for precision, and thus numerical reporting for these would be rather imprecise. pH was controlled during all experiments through carbonate buffering, and remained within a narrow range (7.48-7.75). At this pH range, ammonia was measured below detection; also, nitrite was measured below detection, suggesting complete nitrification processes. As then reported in our manuscript, ammonium and nitrate concentrations were measured via photometer as the dependent variables reported in our results. We have added clarification in two places in the Methods, including the statement (lines 228-231, Manuscript Markup):

All parameters remained consistent within broad ranges indicated by test strips throughout the inoculation period. The pH remained within a constrained range (7.48-7.75) because of carbonate buffering; as such, ammonia and nitrite were below detection throughout inoculation and experiment periods.

And the later statement amended as such (lines 253-256, Manuscript Markup):

The dissolved oxygen concentration, pH, and temperature measurements were obtained both at the beginning and at the end of every experimental trial, with pH remaining within the prescribed range (7.48-7.75), dissolved oxygen controlled (4.9-5.6 ppm), and temperature constrained (30°C).

• Line 303: it cannot be claimed that SSA is the main parameter for maximizing the performance of MBBRs as many other factors influence the whole treatment process.

Indeed, the reviewer is correct in that the text can be misconstrued as currently expressed. We thank the reviewer for pointing this out. We have replaced this statement and sharpened the language for clarification. The revised statements now suggest a more nuanced meaning of our interpretation (lines 347-350, Manuscript Markup):

The observed variation in the ammonia removal performance when manipulating the SSA parameter reinforces previously reported findings (36) that SSA is an important parameter for MBBR design. Furthermore, these findings strongly suggest that SSA can be used as a proxy parameter when seeking to optimize MBBR performance.

• Ammonia removal close to 100% might be due to the adaption of microbes at low concentration of ammonia being fed in the bioreactor- the case might be opposite with real field wastewater or when the concentration of ammonia is more than 5 mg/L.

Thank you for your statement. We agree with your interpretation of the processes occurring. Indeed, it is possible that the microbes are well-adapted to the comparatively lower ammonia concentrations in the wastewater simulant. However, it should be noted that the ammonia concentrations are scaled for a particular wastewater from an aquaculture system, where ammonia concentrations are typically lower than that found in municipal wastewater because of the particular dilutions. Results on performance of the biofilter system and media may indeed be different for higher concentrations of ammonia in the input wastewater. We support the idea that these sorts of studies should be done in the future to expand the knowledge of performance for these particular biofilter media in other applications. As noted previously, an indication of the need for future research using higher wastewater concentrations is inserted in the Discussion (lines 474-478, Manuscript Markup):

Additionally, performance of the biocarriers should be investigated for applications to wastewater streams, including municipal or aquaculture sources, where higher concentrations in organic carbon and nutrient concentrations differentially impact the accumulation of biofilm on the carriers themselves, and ultimately the performance of the bioreactor in any given application.

• Any relation found for nitrate production and TAN removal, as nitrate production is almost the same whereas trend for TAN removal is quite opposite.

Thank you for your comment. Indeed, removal of TAN and production of nitrate is expected to be nearly equivalent amounts, according to common understanding of the nitrification process. As such, we believe that is reflected in our data as presented, where, in all cases and treatments, nitrate concentration increases proportionally to the decrease in TAN. 

Reviewer #2: 

Abstract: Some data must be provided to support your statements.

Thank you for your comment. We agree that more precise language and presentation of data in the Abstract is best practice for support of our statements. We have rewritten portions of the Abstract to have more detail with support from data, found in the associated lines (lines 31-40 Manuscript Markup):

Results showed that large and medium SSA gyroid biocarriers delivered the best ammonia removal performance of all designs, and significantly better than that of a standard commercial design. The percentage of ammonia nitrogen conversion at 8 hours for the best performing biocarrier design was: 99.33% (large SSA gyroid, constant fill ratio), 94.74% (medium SSA gyroid, constant total surface area), and 92.73% (large SSA gyroid, constant biocarrier media count). Additionally, it is shown that the ammonia conversion performance was correlated to the specific surface area of the biocarrier, with the greatest rates of ammonia conversion (99.33%) and nitrate production (2.7 mg/L) for manufactured gyroid biocarriers with a specific surface area greater than 1980.5 m2/m3.

Page 3, lines 41-44: Introduce the following current reference regarding the applications of MBBR in order to complement the literature cited: “Moving bed biofilm reactor as an alternative wastewater treatment process for nutrient removal and recovery in the circular economy model”.(https://doi.org/10.1016/j.biortech.2019.122631).

Thank you for pointing us to this article. We have added a reference to the paper (line 49-50, Manuscript Markup).

Page 3, line 53: Include the following reference in relation to the filling ratio: “Effect of the filling ratio, MLSS, hydraulic retention time, and temperature on the behavior of the hybrid biomass in a hybrid moving bed membrane bioreactor plant to treat urban wastewater”. (10.1061/(ASCE)EE.1943-7870.0000939)”.

Thank you for your comment. We have added a reference to the paper (line 59, Manuscript Markup).

Page 3, line 60: Correct “MMBR”.

We are grateful for letting us know. We have corrected the text (line 66, Manuscript Markup).

Page 6, line 128: you mention reference 23, but in introduction you say that this kind of design had not been carried out previously for MBBR. Could you justify this contradictory information?

Thank you for your comment. That particular reference demonstrated preliminary work on the production of the gyroid production, but did not explore the performance aspects in a replicated experimental design. We have clarified the language in various places to reflect the preliminary nature of that work, and the follow-on nature of this manuscript, such as in the introductory text (lines 105-108, Manuscript Markup):

In preliminary work on the topic, Elliott et al. (23) demonstrated the manufacture of 3D-printed gyroid-based biocarriers using material jetting technologies and established the feasibility of implementing such technology for nitrification MBBRs.

Likewise, in the line following (line 109-111, Manuscript Markup):

To date, very little research has focused on the evaluation of novel 3D printed carriers in MBBRs to remove ammonia from wastewater and evaluate removal rates as a function of the SSA or the fill ratio of biocarrier media.

Figure 1: Explain why you introduce this figure in the paper.

Thank you for your comment. Based upon your comment, we have reconsidered the information best presented in Figure 1. We have now presented images of each of the four types of biocarrier designs tested in this experiment. The cross-reference for Figure 1 is now rewritten as follows (lines 158-159, Manuscript Markup):

Images of example specimens of all biocarrier designs examined in this research are shown in Figure 1.

Page 7, line 154: Why do you indicate “Table 1” in bold?

Thank you for your comment. Bold-face labels of this type were a legacy of a previous formatting style. We have removed the bold type from all cross-reference labels of this type globally. 

Table 1: some lines of the table are missing. Specify the meaning of SSA and include the abbreviation in brackets. Indicate the reference where you have obtained the value of SSA and density for K1 carrier from. The meaning of “work envelope” must be explained.

Thank you for your comments. We have reformatted the table to address the problems with lines. We have spelled out “Specific Surface Area (SSA)” in the table headings. We have added notes on the table referencing the source for values for SSA (calculated using design rendering and mesh models). Also, we have added an explanation of the work envelope in the text prior to the table (lines 146-149, Manuscript Markup):

The work envelope is the minimum closed surface that encompasses the entire geometry of the biocarrier. For the gyroid-based topologies used in this study, it consists of the minimum sphere that would theoretically wrap or enclose the entire biocarrier. For the Kaldnes K1 topology, it consists of the minimum cylinder that would encompass the entire biocarrier.

Figures 2 - 6: They have not enough quality. Please improve them.

Thank you for your comment. We have included high resolution files of all figures now for improving.

Page 8, line 174: 5 ppm for dissolved oxygen is excessive.

Thank you for your comment. We chose a concentration of 5 mg/l of dissolved oxygen was due to the intention to test TAN removal rates at their maximum performance level, and the way to gain control on this parameter was to avoid potential scenarios of dissolved oxygen limitation, as referred to in literature. We have added an explanation of this with a citation as follows (lines 186-189, Manuscript Markup):

To maintain a concentration of dissolved oxygen adequate for aerobic conditions to support nitrification (5 ppm, as in (9)), the air supply pressure for the bioreactors was kept constantly at an operational level of 1 psi, monitored with an analog pressure gage.

Section “Wastewater media preparation” is too long. You must divide it into more sections. Furthermore, some descriptions of analytical methods can be summarized or included in Supplementary Information.

Thank you for your comments. We have divided the entire “Methods’ section into more subsections, to include the following: Biocarrier Design and Fabrication; Biofilter Reactor Design and Setup; Wastewater Media Preparation and Analysis; Experimental Design and Analysis of Results. We believe this organizes better the entire section for clarity of presentation.

Table 2: remove it and indicate the volume in the text (since the concentrations are given).

Thank you for your comment. We have removed Table 2, and indicated the media recipe in detail in the text, as follows (lines 204-207, Manuscript Markup):

The media recipe involved 17L of dechlorinated water, into which was dissolved ammonium chloride (170 mg), calcium carbonate (340 mg) and sodium bicarbonate (595 mg), plus trace elements necessary for nitrification provided by a low concentration (0.2% solution, 34 ml) marine salt solution (Seachem, Inc., Madison, Ga).

Page 10, line 221: What does “uL” mean?

Thank you for your comment. That abbreviation is supposed to represent ‘micro-liters’. We have edited it to reflect the proper symbol (line 236, Manuscript Markup).

Page 12, line 267: Justify the choice of 0.055 m2. It is not clear in the text.

Thank you for your comment. We determined this surface area through preliminary investigations not reported here. To that, we have attempted to clarify the writing in this section, which now reads as follows (lines 296-299, Manuscript Markup):

The total surface area per reactor of 0.055 m2 was chosen based on the minimum amount of surface area for gyroids needed for a measurable rate of nitrification through the trial, as determined through preliminary investigation and confirmed repeated in other experiments (for example, Experiment 1, Table 2).

We have also added clarification of the information determined from preliminary investigation earlier in the Methods section, with the following statements (lines 278-283, Manuscript Markup):

In all experiments, sizing was determined based on results from preliminary trials (not reported here). From preliminary trials it was identified that 2.00x10-04 m3 was a workable volume of packed biocarriers (for all types) that provided an observable mobility inside the reactor. Such volume selection consequently derived into the selection of the number of each carrier of each type for each of the experimental treatments described below.

Table 4: Explain the meaning of “packing volume” in the text in order to understand the information supplied in this table.

Thank you for your comment. We have made changes to the text in an attempt to clarify, including the following (lines 266-269, Manuscript Markup):

The packing volume, which is used to estimate the fill ratio, refers to the volume occupied by a predetermined count of biocarrier elements and is dependent on the overall geometry and surface topology nesting characteristics, which would allow for a denser packing arrangement.

Page 12, line 276: justify with some reference the number of 7 carriers.

Thank you for your comment. We determined the sufficiency of 7 carriers through preliminary investigations that are not reported here. We have added this clarification in the text as follows (lines 307-310, Manuscript Markup):

The constant count value of 7 units of media available in each reactor was chosen based on the minimum number of Kaldnes K1 biocarriers needed for a detectable amount of total ammonia conversion during the 8-hour trials, as determined through observations in preliminary investigations (data not shown).

Section “Results and Discussion” must be divided in several sub-sections according to the three experiments carried out.

Thank you for your comment and suggestion. We have now added the following subsection headings: “Experiment 1: Constant Fill Ratio of Biocarriers” (line 337, Manuscript Markup); “Experiment 2: Constant Biocarrier Total Surface Area” (line 404, Manuscript Markup); “Experiment 3: Constant Biocarrier Count” (line 433, Manuscript Markup); “Summary of All Results” (line 462, Manuscript Markup).

Page 13, line 300: do not use capital letters and use “total ammonia nitrogen” without “-N”.

Thank you for your comment. We have corrected the terminology in that location and elsewhere when applicable. 

Page 13, line 302: it is not logical to give a p-value with three significant numbers.

Thank you very much for pointing this out. We revised the accepted format for the levels of p-values and have changed it in multiple places to the journal accepted notation of p<0.001 (instead of p=0.000).

Revise p-values in manuscript and introduce the statistical method in “Materials and Methods”.

Thank you for your comment. We have revised p-values globally through the document. Additionally, we have added a subsection heading in the Methods section entitled “Experimental Design and Analysis of Results” (line 259, Manuscript Markup). We have added the following explanation of statistical analyses in this section (lines 320-325, Manuscript Markup):

An analysis of homoscedasticity and a Kolmogorov-Smirnov normality test showed that the data do not follow the underlying assumptions an Analysis of Variance. Consequently, the statistical analysis was conducted with a non-parametric Kruskal-Wallis statistical test on the medians. Additional analysis was performed with a Mood’s Median test to compare the medians of pairwise samples to determine significance.

I can not revise the results from Figure 4 as I can not see them (poor quality of the figure).

Thank you for your comment. We have revised Figure 4 and reproduced it at a high resolution for quality. 

Please, do not use abbreviations in X or Y axis.

Thank you for your comment. We have revised all figures to remove abbreviations for clarity.

When you say “ammonia removed”, what do you refer to? In your systems, there is no an anoxic zone or anoxic time. Explain it better in order to get a better comprehension for potential readers. I suppose that you consider the transformation of ammonium nitrogen into nitrate nitrogen (nitrification only). Specify clearly it (“remove” is confusing). Additionally, explain why you analyze both parameters TAN and nitrate concentration.

Thank you for your comment. We agree that the language can be confusing, as indeed there is no removal of total nitrogen, as might occur if anoxic zones are present to allow for possible denitrification. Rather, we are discussing conversion of ammonia nitrogen to nitrate nitrogen through the well-known process of nitrification, which we are measuring indirectly through the expected decrease in ammonia concentrations and proportionate increase in nitrate concentrations. To that, we have amended the language in various locations throughout, especially accentuating that the process being described is nitrification as represented by the conversion of ammonia nitrogen to nitrate nitrogen (for example, line 51, Manuscript Markup; also, line 264, Manuscript Markup), and by the replacement of the word “removal” with the words “ ammonia nitrogen conversion”, globally in multiple places distributed throughout the text.

Page 15, lines 345-349: compare with the values of protected surface area for biofilm growth corresponding to your biocarriers.

Thank you for this comment. The protected area of a biocarrier is not a design parameter but rather a limitation of the design as expressed by the Kaldnes K1 manufacturer. There is no unique definition of protected area but it is loosely used in commercial terminology to explain why biofilm develops only on selected areas. Because this term is not well defined, and also because it is not critical to any discussion in the manuscript, we have removed all verbiage regarding protected surface areas from the manuscript.

Figure 6: it is not easily interpretable. Moreover, use superscripts.

Thank you for your comments. We have recrafted Figure 6 into two separate panels to better demonstrate the relationships. Also, we have corrected the superscripts in the figure. 

Page 16, lines 394-395: Justify what you say.

Thank you for your comment. We believe that this statement is redundant with the meaning of the previous sentence, where the performance results of each of the biocarrier types is discussed in detail. Because of this, we have chosen to remove this sentence.

Pages 16-17, lines 396-400: Explain it with more detail.

Thank you for your comments. We have attempted to clarify our writing here with more detail. It now reads as follows (lines 455-461, Manuscript Markup):

The TAN percentage change from gyroids was measurably greater than conventional biocarriers even when the biocarrier count was notably less than that compared to other experiments. This suggests that the performance of the gyroid biocarriers is enhanced by the high specific surface area. In addition, the amount of increases in performance for the medium and large SSA gyroids, despite the relatively number of biocarriers in this experiment, suggests a subsidy in performance resulting from the high proportion of protected surface area, a result directly related to the conformation of the topographical design of the biocarriers.

Conclusions: this section must be summarized, you must give conclusions without repeating information from Materials and Methods for example. You can number the main conclusions for each experiment carried out. Include some data to support your conclusions. Explain the last paragraph: you state 1013 m2/m3 as the most suitable carrier, which carrier does this value correspond to? Justify this last paragraph.

Thank you for your comment. We have rewritten the Conclusions based upon your recommendations. The Conclusions section now reads as follows (lines 483-505, Manuscript Markup):

The levels of geometrical complexity of the design and fabrication of biocarriers for water treatment processes were manipulated with the capabilities of additive manufacturing. Three mathematical designs with increasing specific surface area (SSA) were used to fabricate biocarriers and test their performance three different scenarios: constant fill ratio, constant total surface area, and constant biocarrier media count. For all the scenarios, the gyroid media designs presenting large and medium SSA provide the best nitrification performances: that is, the lowest final concentration of pooled total ammonia nitrogen as well the highest concentration of pooled nitrate nitrogen. The best performing design for the scenario of constant fill ratio is the large SSA gyroid with 99.33% of ammonia nitrogen conversion at 8 hours. The best performing design for the scenario of constant total surface area is the medium SSA gyroid with 94.74% of ammonia nitrogen conversion at 8 hours. The best performing design for the scenario of constant biocarrier media count is the large SSA gyroid with 92.73% of ammonia nitrogen conversion at 8 hours. From a practitioner’s standpoint, experiment 1 perhaps provides the most relevant information for selection of biofilter media, as one of the most practical considerations in bioreactors loading conditions is the fill ratio in the tanks. In this regard, and with manufacturing considerations aside, the results indicate that either large or medium gyroid SSA would be the best choice. However, this study was focused on the early stages of nitrification and thus limited to an 8-hour period. Therefore, it ignores some of the steady-state operational issues (e.g. clogging, biofilm build-up, sloughing, etc.) of bioreactors that would likely give a functional advantage to a design with a larger pore openings while maintaining optimal performance (i.e. medium SSA gyroid). Additionally, results from experiment 2 and 3 indicate that further optimization on the geometry and its proxy SSA can be pursued. Future work will aim at this through the use of a statistical response surface methodologies over a longer period of bioreactor performance.

---

## [Decision Letter · Decision Letter 1]

27 Jul 2020

PONE-D-20-13545R1

Analysis of very-high surface area 3D-printed media in a moving bed biofilm reactor for wastewater treatment

PLOS ONE

Dear Dr. Blersch,

Thank you for submitting your manuscript to PLOS ONE. After careful consideration, we feel that it has merit but does not fully meet PLOS ONE’s publication criteria as it currently stands. Therefore, we invite you to submit a revised version of the manuscript that addresses the points raised during the review process.

We look forward to receiving your revised manuscript.

Kind regards,

Leonidas Matsakas

Academic Editor

PLOS ONE

Reviewers' comments:

Reviewer's Responses to Questions

**Comments to the Author**

1. If the authors have adequately addressed your comments raised in a previous round of review and you feel that this manuscript is now acceptable for publication, you may indicate that here to bypass the “Comments to the Author” section, enter your conflict of interest statement in the “Confidential to Editor” section, and submit your "Accept" recommendation.

Reviewer #1: All comments have been addressed

Reviewer #2: (No Response)

2. Is the manuscript technically sound, and do the data support the conclusions?

Reviewer #1: Yes

Reviewer #2: Yes

3. Has the statistical analysis been performed appropriately and rigorously? 

Reviewer #1: Yes

Reviewer #2: Yes

4. Have the authors made all data underlying the findings in their manuscript fully available?

Reviewer #1: Yes

Reviewer #2: Yes

5. Is the manuscript presented in an intelligible fashion and written in standard English?

Reviewer #1: Yes

Reviewer #2: Yes

6. Review Comments to the Author

Reviewer #1: Authors have responded to all the comments and Manuscript has been significantly revised with scientific contents.

Reviewer #2: Authors have addressed most of comments. However, figures are presented in different sub-figures (in several pages); it would be more suitable if each figure (with all sub-figures) is included in one page. Table 1 contains several symbols "*" and "**" but the meaning of them is missing. Finally, when you speak about the surface area (0.055 m2) and the numbers of carriers (7) it is necessary to specify the references used (it is not sufficient to say "preliminary investigation").

7. PLOS authors have the option to publish the peer review history of their article (what does this mean?). If published, this will include your full peer review and any attached files.

Reviewer #1: No

Reviewer #2: No

---

## [Author Response · Author response to Decision Letter 1]

12 Aug 2020

We present the response to reviewers in detail in this communication. We thank the reviewers again for their helpful and constructive comments that continue to improve this manuscript. We have addressed all comments in this letter and/or in the manuscript. 

Reviewer #2: 

Authors have addressed most of comments. However, figures are presented in different sub-figures (in several pages); it would be more suitable if each figure (with all sub-figures) is included in one page. 

Thank you for your comments and observations on the figures. We have re-combined all the sub-figures as panels in each overall figure, formatted to fit on one page. 

• Table 1 contains several symbols "*" and "**" but the meaning of them is missing. 

Thank you for catching that error resulting from formatting the table for submission. We have now included the table footnotes that are indicated by those symbols (Table 1, p. 7, markup manuscript).

Finally, when you speak about the surface area (0.055 m2) and the numbers of carriers (7) it is necessary to specify the references used (it is not sufficient to say "preliminary investigation").

Thank you for your comment. The theoretically required surface area was initially calculated using performance metrics estimated from literature, and then validated through trial-and-error experimentation in preliminary work in the laboratory, the data for which are not reportable. However, the calculations that support the initial planning are reported in a published document. We have included this reference in the text, and indicated more clearly the preliminary aspects of the trials, in the following location (lines 278-280, Manuscript Markup):

"In all experiments, sizing was determined based on calculations of minimum required biocarrier surface area for complete nitrification, as reported in [36, p.95] and validated through trial and error in preliminary trials (data not reported here)."

Where the reference that is newly included as a citation (line 629-630, Manuscript Markup):

Proaño Peña GF. 3D-printed Custom Substratum for Fast Functional Responses from Microbial Colonization. Auburn, Alabama, USA: Auburn University; 2018.

Also, in the following location (lines 297-300, Manuscript Markup):

"The total surface area per reactor of 0.055 m2 was chosen based on the minimum amount of surface area for gyroids needed for a measurable rate of nitrification throughout the time of the trial, as determined through trial and error preliminary investigations and confirmed through repetition in other experiments (for example, Experiment 1, Table 2)."

Also, in the following location (line 310-313, Manuscript Markup):

"The constant count value of 7 units of biocarriers in each reactor was chosen based on the minimum number of Kaldnes K1 biocarriers needed for a detectable amount of total ammonia conversion during the 8-hour trials, as determined through observations in trial and error preliminary investigations (data not included here)."

---

## [Editor Report · Decision Letter 2]

17 Aug 2020

Analysis of very-high surface area 3D-printed media in a moving bed biofilm reactor for wastewater treatment

PONE-D-20-13545R2

Dear Dr. Blersch,

We’re pleased to inform you that your manuscript has been judged scientifically suitable for publication and will be formally accepted for publication once it meets all outstanding technical requirements.

Kind regards,

Leonidas Matsakas

Academic Editor

PLOS ONE
---

## [Editor Report · Acceptance letter]

19 Aug 2020

PONE-D-20-13545R2 

Analysis of very-high surface area 3D-printed media in a moving bed biofilm reactor for wastewater treatment 

Dear Dr. Blersch:

I'm pleased to inform you that your manuscript has been deemed suitable for publication in PLOS ONE. Congratulations! Your manuscript is now with our production department. 

Kind regards, 

on behalf of

Dr. Leonidas Matsakas 

Academic Editor

PLOS ONE